# High-risk *Escherichia coli* clones that cause neonatal meningitis and association with recrudescent infection

Nguyen Thi Khanh Nhu[1,2,3], Minh-Duy Phan[1,2,3], Steven J Hancock[2,3†], Kate M Peters[1,2,3], Laura Alvarez-Fraga[2,3‡], Brian M Forde[3,4], Stacey B Andersen[5], Thyl Miliya[6], Patrick NA Harris[4,7], Scott A Beatson[2,3], Sanmarie Schlebusch[4,7,8], Haakon Bergh[7], Paul Turner[6,9], Annelie Brauner[10], Benita Westerlund-Wikström[11], Adam D Irwin[3,4,12]*, Mark A Schembri[1,2,3]*

[1]Institute for Molecular Bioscience (IMB), The University of Queensland, Brisbane, Australia; [2]School of Chemistry and Molecular Biosciences, The University of Queensland, Brisbane, Australia; [3]Australian Infectious Diseases Research Centre, The University of Queensland, Brisbane, Australia; [4]University of Queensland Centre for Clinical Research, The University of Queensland, Brisbane, Australia; [5]Genome Innovation Hub, The University of Queensland, Brisbane, Australia; [6]Cambodia Oxford Medical Research Unit, Angkor Hospital for Children, Siem Reap, Cambodia; [7]Pathology Queensland, Queensland Health, Brisbane, Australia; [8]Q-PHIRE Genomics and Public Health Microbiology, Forensic and Scientific Services, Coopers Plains, Brisbane, Australia; [9]Centre for Tropical Medicine and Global Health, Nuffield Department of Medicine, University of Oxford, Oxford, United Kingdom; [10]Department of Microbiology, Tumor and Cell Biology, Division of Clinical Microbiology, Karolinska Institutet and Karolinska University Hospital, Stockholm, Sweden; [11]Molecular and Integrative Biosciences Research Programme, University of Helsinki, Helsinki, Finland; [12]Infection Management Prevention Service, Queensland Children's Hospital, Brisbane, Australia

*For correspondence:
a.irwin@uq.edu.au (ADI);
m.schembri@uq.edu.au (MAS)

Present address: †Wellcome-Wolfson Institute for Experimental Medicine, School of Medicine, Dentistry and Biomedical Sciences, Queen's University Belfast, Belfast, United Kingdom; ‡INRAE, Univ Montpellier, LBE, Narbonne, France

Competing interest: The authors declare that no competing interests exist.

**Abstract** Neonatal meningitis is a devastating disease associated with high mortality and neurological sequelae. *Escherichia coli* is the second most common cause of neonatal meningitis in full-term infants (herein NMEC) and the most common cause of meningitis in preterm neonates. Here, we investigated the genomic relatedness of a collection of 58 NMEC isolates spanning 1974–2020 and isolated from seven different geographic regions. We show NMEC are comprised of diverse sequence types (STs), with ST95 (34.5%) and ST1193 (15.5%) the most common. No single virulence gene profile was conserved in all isolates; however, genes encoding fimbrial adhesins, iron acquisition systems, the K1 capsule, and O antigen types O18, O75, and O2 were most prevalent. Antibiotic resistance genes occurred infrequently in our collection. We also monitored the infection dynamics in three patients that suffered recrudescent invasive infection caused by the original infecting isolate despite appropriate antibiotic treatment based on antibiogram profile and resistance genotype. These patients exhibited severe gut dysbiosis. In one patient, the causative NMEC isolate was also detected in the fecal flora at the time of the second infection episode and after treatment. Thus, although antibiotics are the standard of care for NMEC treatment, our data suggest that failure to eliminate the causative NMEC that resides intestinally can lead to the existence of a refractory reservoir that may seed recrudescent infection.

## eLife assessment

This **valuable** study presents findings characterising the genomic features of *E. coli* isolated from neonatal meningitis from seven countries, and documents bacterial persistence and reinfection in two case studies. The genomic analyses are **solid**, although the inclusion of a larger number of isolates from more diverse geographies would have strengthened the generalisability of findings. The work will be of interest to people involved in the management of neonatal meningitis patients, and those studying *E. coli* epidemiology, diversity, and pathogenesis.

## Introduction

Neonatal meningitis (NM) is a devastating disease with a mortality rate of 10–15% and severe neurological sequelae including hearing loss, reduced motor skills, and impaired development in 30–50% of cases (*Doctor et al., 2001*; *Stevens et al., 2003*; *Harvey et al., 1999*). The incidence of disease is highest in low-income countries and occurs at a rate of 0.1–6.1/1000 live births (*Harvey et al., 1999*). *Escherichia coli* is the second most common cause of NM in full-term infants (herein NMEC), after group B *Streptococcus* (GBS) (*Ouchenir et al., 2017*; *Gaschignard et al., 2011*), and the most common cause of meningitis in preterm neonates (*Gaschignard et al., 2011*; *Basmaci et al., 2015*). Together, these two pathogens cause ~60% of all cases, with on average one case of NMEC for every two cases of GBS (*May et al., 2005*; *Holt et al., 2001*). In several countries, NM incidence caused by GBS has declined due to maternal intrapartum antibiotic prophylaxis; however, NM incidence caused by *E. coli* remains the same (*May et al., 2005*; *van der Flier, 2021*). Moreover, NMEC is a significant cause of relapsed infections in neonates (*Anderson and Gilbert, 1990*).

NMEC are categorised genetically based on multi-locus sequence type (ST) or by serotyping based on cell-surface O antigen (O), capsule (K), and flagella (H) antigens. Analysis of NMEC diversity in France revealed ~25% of isolates belong to the ST95 clonal complex (STc95) (*Geslain et al., 2019*), however, a global picture of NMEC epidemiology is lacking. NMEC possess a limited diversity of serotypes, dominated by O18:K1:H7, O1:K1, O7:K1, O16:K1, O83:K1, and O45:K1:H7, which together account for >70% of NMEC (*Sarff et al., 1975*; *Plainvert et al., 2007*; *Bidet et al., 2007*; *Johnson et al., 2002*). Notably, ~80% of NMEC express the K1 capsule, a polysaccharide comprising linear homopolymers of α2–8-linked *N*-acetyl neuraminic acid (*Sarff et al., 1975*; *Robbins et al., 1974*). Apart from the K1 capsule, specific NMEC virulence factors are less-well defined, though studies have demonstrated a role for S fimbriae (*Prasadarao et al., 1993*), the outer membrane protein OmpA (*Prasadarao et al., 1996*), the endothelial invasin IbeA (*Huang et al., 2001*), and the cytotoxin necrotising factor CNF1 (*Wang and Kim, 2013*) in translocation of NMEC across the blood–brain barrier and pathogenesis. A large plasmid encoding colicin V (ColV), colicin Ia bacteriocins, and several virulence genes including iron-chelating siderophore systems has also been strongly associated with NMEC virulence (*Peigne et al., 2009*).

Despite being the second major NM aetiology, genomic studies on NMEC are lacking, with most reporting single NMEC complete genomes. Here, we present the genomic analyses of a collection of 58 NMEC isolates obtained from seven different geographic regions over 46 years to understand virulence gene content, antibiotic resistance, and genomic diversity. In addition, we provide a complete genome for 18 NMEC isolates representing different STs, serotypes, and virulence gene profiles, thus more than tripling the number of available NMEC genomes that can be used as references in future studies. Three infants in our study suffered recrudescent invasive NMEC infection, and we show this was caused by the same isolate. We further revealed that patients that suffered recrudescent invasive infection had severe gut dysbiosis, and detected the infecting isolate in the intestinal microflora, suggesting NMEC colonisation of the gut provides a reservoir that can seed repeat infection.

## Results

### Establishment of an NMEC collection from geographically diverse locations

A collection of 52 NMEC isolates cultured from the blood or cerebrospinal fluid (CSF) of neonates with meningitis was established with the addition of six completely sequenced NMEC genomes available

on the NCBI database. This yielded a final set of 58 NMEC isolates spanning 1974–2020. The isolates were obtained from seven different geographic locations; Finland (*n* = 17, 29.3%), Sweden (*n* = 14, 24.1%), Australia (*n* = 15, 25.9%), Cambodia (*n* = 7, 12.1%), USA (*n* = 3, 5.2%), France (*n* = 1, 1.7%), and the Netherlands (*n* = 1, 1.7%).

## ST95 and ST1193 are the two major STs of NMEC

Phylogenetic analysis was performed on the 58 NMEC isolates, with an additional eight well-characterised *E. coli* strains included for referencing (EC958, UTI89, MS7163, CFT073, UMN026, 536, APEC01, and MG1655). The NMEC isolates were diverse, and spanned phylogroups A, B2, C, D, and F; the majority of isolates were from phylogroup B2 (*n* = 48, 82.8%). Overall, the isolates belonged to 22 STs, of which 15 STs only contained one isolate. ST95 (*n* = 20, 34.5%) and ST1193 (*n* = 9, 15.5%) were the two most common NMEC STs (*Figure 1*, *Supplementary file 1*). ST95 isolates were obtained over the entire study period, while ST1193 isolates were more recent and only obtained from 2013. Four isolates belonged to ST390 (6.9%), which is part of the STc95. One isolate belonged to a novel ST designated ST11637, which is part of the ST14 clonal complex (STc14) that also contains ST1193 (*Figure 1*, *Supplementary file 1*). Isolates from other common phylogroup B2 extra-intestinal pathogenic *E. coli* (ExPEC) lineages, ST131, ST73, and ST69, as well as several STs associated with environmental sources such as ST48 and ST23, were detected in the collection. However, it is notable that the high incidence of NM associated with ST95 and ST1193 does not reflect the broader high prevalence of major ExPEC clones associated with human infections in the publicly available Enterobase database (*Zhou et al., 2020*; *Figure 1—figure supplement 1*), suggesting ST95 and ST1193 exhibit specific virulence features associated with their capacity to cause NM.

Eighteen NMEC isolates were completely sequenced using complementary long-read Oxford Nanopore Technology, enabling accurate comparison of NMEC genome size, genomic island composition and location, and plasmid and prophage diversity. These isolates spanned the diversity in the collection, representing 11 different STs, including two ST1193 isolates (one with the dominant O75:H5 serotype and one with an unusual O6:H5 serotype), five ST95 isolates with different serotypes, and one isolate from the novel ST11637.

## Antibiotic resistance in NMEC

Antibiotic resistance profiling revealed an overall low level of resistance in the collection. The ST1193 isolates contained fluoroquinolone resistance defining mutations in *gyrA* (S83L D87N) and *parC* (S80I), as previously described for this lineage (*Johnson et al., 2019*). In addition, 77.8% of ST1193 isolates (7/9 isolates) also harboured at least one gene conferring resistance to aminoglycosides (*aac(3)-IId*, *aadA5*, *aph(3″)-Ib*, and *aph(6)-Id*), trimethoprim (*dftA17*), and sulphonamides (*sul1* and *sul2*) (*Figure 1—figure supplement 2*). Six out of the seven isolates from Cambodia had more than one antibiotic resistance gene, likely reflecting increased antibiotic resistance rates in this region (*Reed et al., 2019*). Indeed, in addition to *gyrA* and *parC* mutations for fluoroquinolone resistance, CAM-NMEC-6 contains 14 antibiotic resistance genes (including resistance to third-generation cephalosporins and carbapenems) and CAM-NMEC-4 contains 11 antibiotic resistance genes (*Figure 1—figure supplement 2*).

## Virulence factors in NMEC

The isolates exhibited variable distribution of virulence genes previously linked to NMEC pathogenesis. The most prevalent genes were those involved in iron uptake, including the enterobactin (98%), yersiniabactin (98%), aerobactin (62%), and salmochelin (55%) siderophore systems, and the heme receptors *chuA* (93%) and *hma* (62%) (*Figure 1*). Also common were the *sitABCD* genes encoding an iron/manganese transporter (98%). The presence of fimbrial and afimbrial adhesins was also diverse. The most prevalent adhesins were type 1 fimbriae (100%), *mat* (*ecp*) fimbriae (98%), and the *fdeC* adhesin (98%). Genes encoding P and S fimbriae were detected in 36% and 22% of NMEC isolates, respectively. The most prevalent toxin was the uropathogenic-specific genotoxin *usp* (83%), which was only found in phylogroup B2 isolates. Other toxin genes encoding the serine protease autotransporters Vat (65% prevalence) and Sat (29%), hemolysin (12%), and cytotoxic necrotising factor-1 (7%) were less prevalent. Additional virulence genes included the *aslA* arylsulfatase (95%), the *iss* lipoprotein (76%), and the *ibeA* invasin (33%). The ColV-plasmid was present in 33% of the isolates (*Figure 1*, *Supplementary file 1*). Direct comparison of virulence factors between ST95 and ST1193, the two

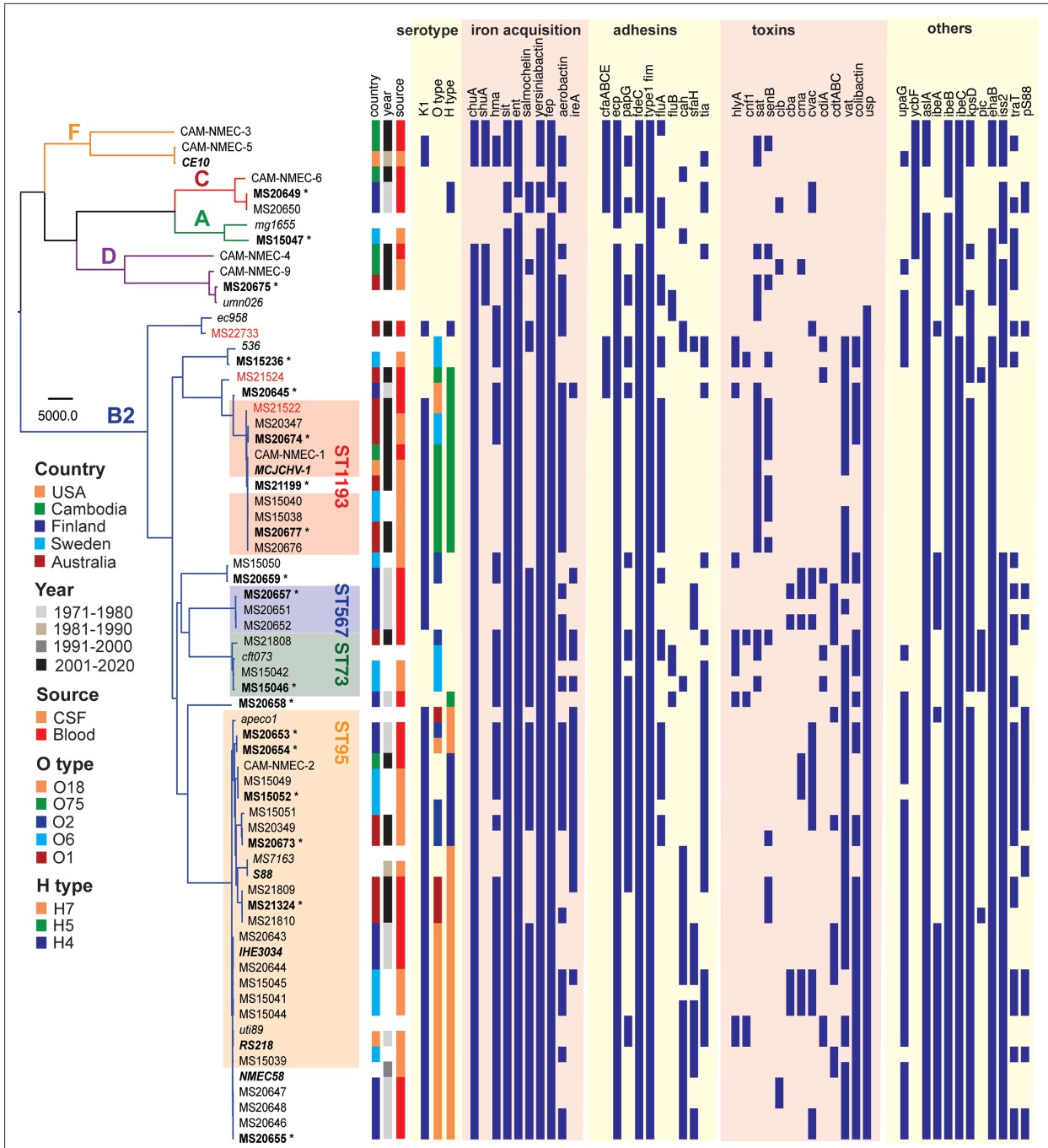

**Figure 1.** Maximum likelihood phylogram displaying the relationship of the NMEC isolates with their associated serotype and virulence factor profile. Non-NMEC isolates used in the analysis for referencing are italicised. The phylogram was built and recombination regions removed employing Parsnp, using 185,911 core single-nucleotide polymorphisms (SNPs) and NMEC strain IHE3034 as the reference. The scale bar indicates branch lengths in numbers of SNPs. NMEC isolates with available complete genomes are bold-italicised, while NMEC isolates that were completely sequenced in this study are indicated in bold and marked with an asterisk. The NMEC isolates that caused recrudescent invasive infection in this study are indicated in red. Branches are coloured according to phylogroups: orange, phylogroup F; red, phylogroup C; green, phylogroup A; violet, phylogroup D; and blue, phylogroup B2. The presence of specific virulence factors is indicated in dark blue. The phylogeny can be viewed interactively at https://microreact.org/project/oNfA4v16h3tQbqREoYtCXj-high-risk-escherichia-coli-clones-that-cause-neonatal-meningitis.

The online version of this article includes the following figure supplement(s) for figure 1:

**Figure supplement 1.** Number of human-derived *E. coli* strains from ST95, ST1193, ST38, ST131, ST73, ST10, and ST69 available in the Enterobase database.

*Figure 1 continued on next page*

*Figure 1 continued*

**Figure supplement 2.** Antibiotic resistance gene profile of NMEC strains in the collection.

**Figure supplement 3.** ST95 NMEC strains contain more virulence factors than ST1193 NMEC strains.

**Figure supplement 4.** K1 capsule production in NMEC.

most dominant NMEC STs, revealed that the ST95 isolates ($n$ = 20) contained significantly more virulence factors than the ST1193 isolates ($n$ = 9); p-value <0.001, Mann–Whitney two-tailed unpaired test (*Figure 1—figure supplement 3*).

## NMEC comprise a dominant K1 capsule type and a limited pool of O and H serotypes

The capsule type of the NMEC isolates was determined by in silico typing. K1 was the dominant capsule type in the collection (43/58 isolates, 74.1%) (*Figure 1*). Thirty-four of these isolates were available for capsule testing, and we confirmed K1 expression by ELISA in all but two isolates (*Figure 1—figure supplement 4*). Other capsule types included K2, K5, and K14 (*Supplementary file 1*). A capsule type could not be resolved for 12 isolates, of which eight did not possess a Group II or III capsule type based on the absence of the conserved *kpsD* gene (*Figure 1*; *Supplementary file 1*).

In silico O antigen (O) and flagella (H) serotypes were also determined. O18 was the most common O type ($n$ = 16, 27.6%), followed by O75 ($n$ = 8, 13.9%) and O2 ($n$ = 7, 12.1%). The most dominant H types were H7 ($n$ = 19, 32.8%), H5 ($n$ = 13, 22.4%), and H4 ($n$ = 9, 15.5%). The most common serotype was O18:H7:K1 ($n$ = 14, 24.1%); these isolates belonged to STc95 (nine ST95, four ST390, and one ST416). The second most common serotype was O75:H5:K1 ($n$ = 8, 13.8%); six isolates from ST1193 possessed this serotype.

## NMEC can cause recrudescent invasive infection despite appropriate antibiotic treatment

During 2019 - 2020, three patients from which NMEC isolates were originally cultured suffered recrudescent invasive infection (*Figure 1*; MS21522, MS21524, and MS22733), providing an opportunity to compare the infecting isolates over time using whole-genome sequencing. In all cases, the infecting *E. coli* isolates were susceptible to the therapy, which comprised cefotaxime (50 mg/kg/dose 8 hourly), switched to ceftriaxone (100 mg/kg/day) to facilitate home parenteral antibiotic administration. Bacterial culture was performed from blood, CSF, urine, and/or stool during the infection period (*Figure 2*). These patients were from different regions in Australia.

### Patient 1

Patient 1 (0–8 weeks of age) was admitted to the emergency department with fever, respiratory distress, and sepsis. The child was diagnosed with meningitis based on a CSF pleocytosis (>2000 white blood cells [WBCs], low glucose, elevated protein), positive CSF *E. coli* PCR and a positive blood culture for *E. coli* (MS21522). Two weeks after completion of a 3-week course of appropriately dosed therapy with third-generation cephalosporins as described above, the child developed similar symptoms of fever and irritability. Lumbar puncture was performed and the CSF culture was positive for *E. coli* (MS21576). Both the initial blood culture isolate and the relapse CSF isolate were non-susceptible to ciprofloxacin and gentamycin, and whole-genome sequencing revealed they were identical (ST1193 O18:K1:H5; *fimH*64), with no single-nucleotide polymorphisms (SNPs) nor indels (*Figure 2A*). Unlike the typical ST1193 O75 serotype (*Johnson et al., 2019*), this isolate contained a unique O18 serotype. The isolate possessed mutations in *gyrA* (S83L D87N) and *parC* (S80I), which explain its resistance to ciprofloxacin, as well as a multidrug resistance IncF plasmid containing genes conferring resistance to aminoglycosides (*aac(3)-IId*, *aadA5*, *aph(3")-Ib*, and *aph(6)-Id*), trimethoprim (*dfrA17*), sulphonamides (*sul1* and *sul2*), and macrolides (*mphA*) (*Figure 1—figure supplement 2*). Treatment of the relapse was extended to 6 weeks of intravenous therapy. At follow-up, no anatomical or immunological abnormality was identified and development is normal.

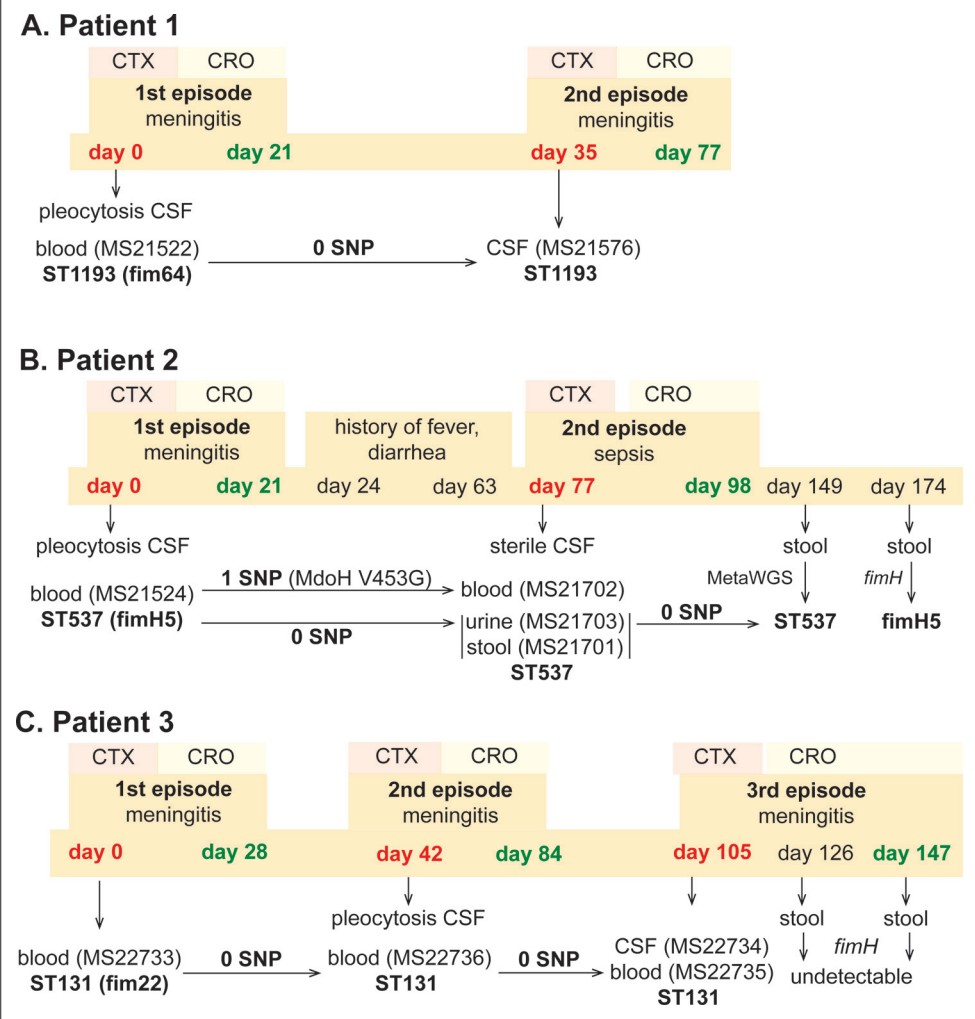

**Figure 2.** Infection and treatment profile of patients suffering NM and recrudescent invasive infection. Indicated is the hospital admission history of patients, together with the timeline of sample collection, identified *E. coli* isolates and their infection source, and isolate identification based on whole-genome sequencing, metagenomic sequencing (MetaWGS), or *fimH* amplicon sequencing. Genomic relatedness is indicated based on the number of single-nucleotide polymorphisms (SNPs). The time of admission for the initial episode is indicated as day 0, with subsequent timepoints indicated as days post initial admission. Admission and discharge days are indicated in red and green, respectively.

## Patient 2

Patient 2 (0–8 weeks of age) presented to the emergency department with a febrile illness. Blood and urine cultures on admission were positive for *E. coli*. CSF taken 24 hr after treatment revealed pleocytosis (>300 WBCs, >95% polymorphs) but no bacteria were cultured. The patient completed a 3-week course of appropriately dosed antibiotic therapy with third-generation cephalosporins. In the 6-week period after discharge, the child had several short admissions to hospital, but no infection was identified. At 11 weeks post initial infection, the child was readmitted to hospital with high fever. CSF cultures were negative and microscopy was normal, but cultures from blood, urine, and faeces were all positive for *E. coli*. Whole-genome sequencing revealed that all isolates belonged to ST537 O75:H5 (*fimH*5; STc14). Pairwise comparison of the recrudescent isolates showed that the urine and fecal isolates were identical to the original isolate, while the blood isolate contained one nonsynonymous SNP in the *mdoH* gene encoding a glucan biosynthesis glucosyltransferase (T1358G; V453G). This mutation is located in the large cytoplasmic domain of MdoH likely involved in polymerisation of glucose from UDP glucose; the isolate exhibited a mucoid colony morphology suggestive of increased colanic acid production. The isolates did not possess plasmids nor antibiotic resistance genes. The

infant experienced recurrent urinary tract infections with *E. coli* and other urinary pathogens through infancy despite normal urinary tract anatomy. At follow-up, no other history of invasive infection nor identified immunodeficiency were noted, and the child was reported to be developing normally.

### Patient 3

Patient 3 (0–8 weeks of age) was admitted to the paediatric intensive care unit with fever and seizures. CSF and blood cultured a fully susceptible *E. coli*. Two weeks after completing a 4-week course of appropriate therapy with third-generation cephalosporins, the infant was readmitted to hospital with fever and irritability, with further investigation identifying *E. coli* in CSF, urine, and blood. Three weeks after the completion of the 6-week treatment course, the infant experienced a second relapse, with *E. coli* isolated from both CSF and blood. Whole-genome sequencing revealed that all isolates were identical and belonged to ST131 O25b:K1:H4 (*fimH*22). These isolates contained a ColV-virulence plasmid, but did not harbour acquired antibiotic resistance genes. The infant received a further 6-week course of therapy. Extensive imaging studies including repeated magnetic resonance imaging of the head and spine, computed tomography imaging of the head and chest, ultrasound imaging of abdomen and pelvis, and nuclear medicine imaging did not show a congenital malformation or abscess. Immunological work-up did not show a known primary immunodeficiency. At 2 years of age, speech delay is reported but no other developmental abnormality.

### The gut as a reservoir to seed recrudescent infection

In all three patients that suffered NM and recrudescent invasive infection, the causative isolates were susceptible to third-generation cephalosporins, suggesting the existence of a persistent reservoir that could evade the cidal effect of antibiotic treatment and seed repeat infection. Indeed, the fact that the causative *E. coli* isolate was detected from a fecal sample at the time of the recrudescent infection in patient 2 (day 77 after initial admission), suggests that NMEC could persist in the gut and cause repeat infection, an observation that has also been reported for uropathogenic *E. coli* that cause recurrent urinary tract infection (*Forde et al., 2019*) and acute pyelonephritis in infants (*Tullus et al., 1984*). Therefore, we retrospectively examined available stored fecal samples from patient 2 at

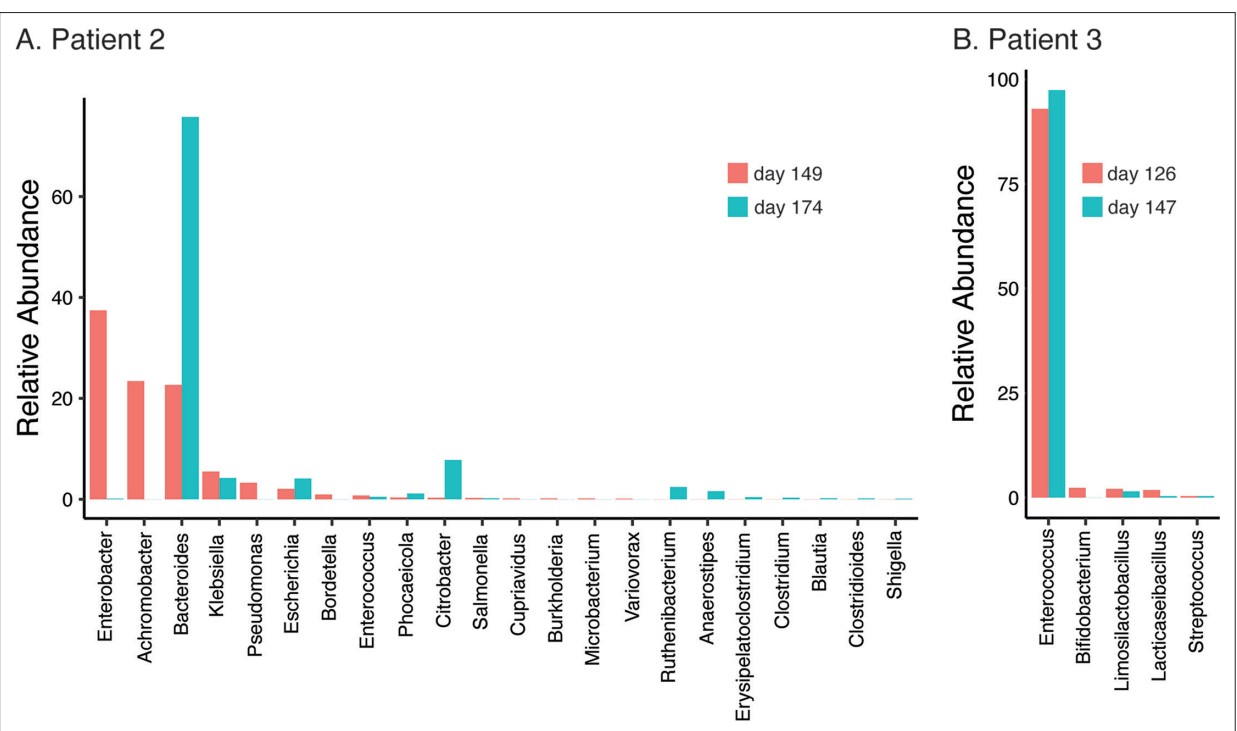

**Figure 3.** Relative abundance of bacterial genera (≥0.01%) in the gut microbiome of patient 2 at 8- and 12-week follow-up post relapsed infection (days 149 and 174 after initial admission) (**A**) and patient 3 during treatment and at discharge after the third episode (days 126 and 147 after initial admission) (**B**).

8- and 12-week follow-up visits post recrudescent infection (days 149 and 174 after initial admission) and patient 3 during treatment and at discharge after the third episode (days 126 and 147 after initial admission) using short-read metagenomic sequencing (*Figure 3*). Although no fecal samples were available for comparative analysis from either patient prior to antibiotic treatment, we observed a low level of diversity in the composition of the microbiome of both patients, consistent with severe dysbiosis. The microbiome of patient 2 was dominated by *Enterobacter* (37.4% relative abundance), *Achromobacter* (23.4% relative abundance), and *Bacteroides* (22.7% relative abundance) genera at 8-week post recrudescent infection (day 149 after initial admission), and by *Bacteroides* genera (75.8% relative abundance) at 12-week post recrudescent infection (day 174 after initial admission). The relative abundance of *E. coli* was 2.05% and 4.1% in each of these samples, respectively, and further analysis using StrainGE (*van Dijk et al., 2022*) showed that the isolates were most closely matched to the original causative MS21524 isolate. We further employed complementary long-read metagenomic sequencing to analyse the 8-week post relapse infection sample, which enabled construction of a complete *E. coli* genome that was identical to the causative ST537 (*fimH*5) isolate (*Figures 2 and 3*; *Supplementary file 3*). In the 12-week post recrudescent infection fecal sample from patient 2, amplicon sequencing targeting *fimH* identified the presence of *E. coli* with the same *fimH* type as the causative isolate (*fimH*5). Thus, two independent analyses of samples taken 4 weeks apart demonstrated the existence of the *E. coli* ST537 isolate in the intestinal microflora of patient 2. In patient 3, the microbiome was dominated by *Enterococcus* genera at both timepoints examined (93% and 97.4% relative abundance). We were unable to detect *E. coli* by *fimH* amplicon sequencing and the relative abundance of *E. coli* in these fecal samples was extremely low (<0.01%) based on metagenomic sequencing (*Supplementary file 3*). The extensive dysbiosis revealed in this patient is likely an outcome of the three rounds of antibiotic treatment.

## Discussion

In this study, we present a genomic analysis of 58 NMEC isolates obtained over 46 years spanning seven different geographic locations and reveal a dominance of ST95 and ST1193. We also provide direct evidence to implicate the gut as a reservoir for recrudescent invasive infection in some patients despite appropriate antibiotic treatment.

The majority of the NMEC isolates in our study belonged to phylogroup B2 (82.8%), an observation consistent with other reports (*Wijetunge et al., 2015*; *Bidet et al., 2007*). These isolates were predominantly from two major STs, ST95, and ST1193. ST95 represents a major clonal lineage responsible for urinary tract and bloodstream infections (*Manges et al., 2019*; *Kallonen et al., 2017*), and were identified throughout the period of investigation. This lineage was also previously shown to cause ~25% of NM cases in France in the period 2004–2015 (*Geslain et al., 2019*), demonstrating its enhanced capacity to cause disseminated infection in newborns. ST1193, on the other hand, was first identified in 2012 (*Platell et al., 2012*), and is the second most common fluoroquinolone-resistant *E. coli* lineage after ST131 (*Johnson et al., 2019*; *Tchesnokova et al., 2019*). ST1193 causing NM was first reported in the USA in 2016 (*Nielsen et al., 2018*). Here, ST1193 accounted for 15.5% of NMEC isolates, all of which were obtained from 2013, and was the dominant lineage since this time. This is consistent with a report in China that showed ST1193 was the most common NMEC (21.4%), followed by ST95 (17.9%), between 2009 and 2015 (*Ding et al., 2021*). Concerningly, the ST1193 isolates examined here carry genes encoding several aminoglycoside-modifying enzymes, generating a resistance profile that may lead to the clinical failure of empiric regimens such as ampicillin and gentamicin, a therapeutic combination used in many settings to treat NM and early-onset sepsis (*Fleiss et al., 2023*; *Fuchs et al., 2018*). This, in combination with reports of co-resistance to third-generation cephalosporins for some ST1193 isolates (*Johnson et al., 2019*; *Ding et al., 2021*), would limit the choice of antibiotic treatments. The dominance of both ST95 and ST1193 in our collection is notable since other widespread *E. coli* phylogroup B2 lineages such as ST131, ST73, ST69, and ST12 do not cause similar rates of NM disease. We speculate this is due to the prevailing K1 polysialic acid capsule serotype found in ST95 and the newly emerged ST1193 clone (*Johnson et al., 2019*; *Goh et al., 2017*) in combination with other virulence factors (*Johnson et al., 2002*; *Wijetunge et al., 2015*; *Bidet et al., 2007*; *Figure 4*) and the immature immune system of preterm infants. Understanding the risk of these clones, as well as perinatal transmission and antibiotic resistance patterns, may inform the appropriateness of interventions such as maternal screening or antimicrobial prophylaxis.

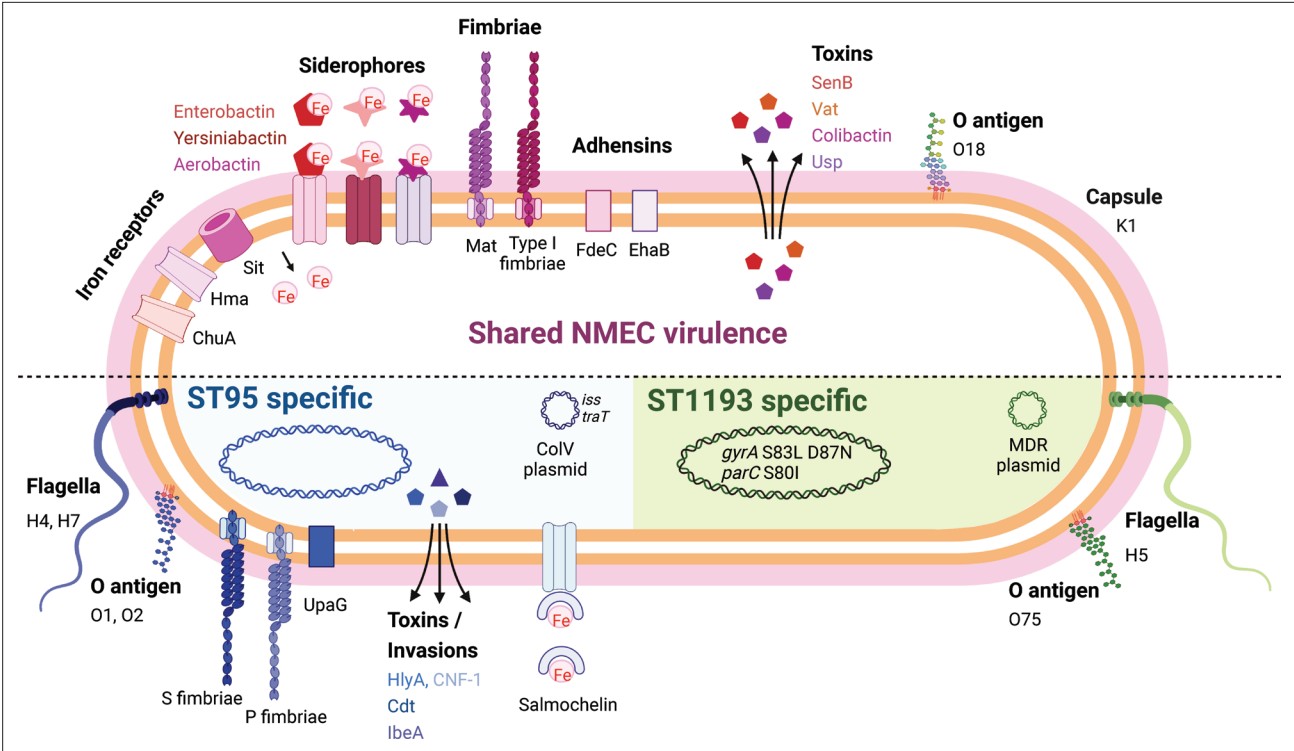

**Figure 4.** Summary of key NMEC virulence genes based on genome profiling performed in this study. Shown are shared virulence genes common to most NMEC, as well as ST95- and ST1193-specific NMEC virulence genes.

Although reported rarely, recrudescent invasive *E. coli* infection in NM patients, including several infants born preterm, has been documented in single study reports (*Vissing et al., 2021*; *Bingen et al., 1993*). In these reports, infants received appropriate antibiotic treatment based on antibiogram profiling and no clear clinical risk factors to explain recrudescence were identified, highlighting our limited understanding of NM aetiology. Here, we tracked NMEC recurrence using whole-genome sequencing in three patients that suffered NM and recrudescent invasive infection, and demonstrated that the isolate causing recrudescence was the same as the original causative isolate and susceptible to the initial antibiotic therapy. In one patient (patient 2), we identified the causative isolate in the stool at days 77, 149, and 174 after initial detection in the bloodstream, providing direct evidence of persistence in the gut, and implicating this site as a reservoir to seed repeat infection. This isolate belonged to ST537 (serotype O75:H5) and is from the same clonal complex as ST1193 (i.e. STc14).

This study had several limitations. First, our NMEC collection was restricted to seven geographic regions, a reflection of the difficulty in acquiring isolates causing this disease. Second, we did not have access to a complete set of stool samples spanning pre- and post-treatment in the patients that suffered NM and recrudescent invasive infection. This impacted our capacity to monitor *E. coli* persistence and evaluate the effect of antibiotic treatment on changes in the microbiome over time. Third, we did not have access to urine or stool samples from the mother of the infants that suffered recrudescence, and thus cannot rule out mother-to-child transmission as a mechanism of reinfection. Finally, we did not have clinical data on the weeks of gestation for all patients, and thus could not compare virulence factors from NMEC isolated from preterm versus term infants. Regardless, our study describes the genomic diversity of NMEC, highlighting ST95 and ST1193 as the most important clonal lineages associated with this devastating disease. Although antibiotics are the standard of care for NMEC treatment, we show that even when appropriate antibiotics are used, in some cases they do not eliminate the causative NMEC that resides intestinally. Together with associated antibiotic-driven dysbiosis, this reveals a need to consider diagnostic and therapeutic interventions to mitigate the risk of recrudescent infection.

## Methods

### Bacterial isolates

A collection of 52 NMEC isolates obtained from 1974 to 2020 was achieved from Sweden, Finland, Cambodia, and Australia. Isolates were stored in glycerol at −80°C until use. All isolates were cultured in Lysogeny broth. The collection comprised 42 isolates from confirmed meningitis cases (29 cultured from CSF and 13 cultured from blood) and 10 isolates from clinically diagnosed meningitis cases (all cultured from blood) (*Supplementary file 1*). This collection was complemented by the addition of six completely sequenced NMEC genomes available on the NCBI database, namely strains IHE3034, RS218, S88, NMEC58, MCJCHV-1, and CE10.

### DNA extraction, genome sequencing, and analyses

Genome sequencing was performed using paired-end Illumina methodology. Illumina sequencing data were processed by removing adapters and low-quality reads using Trimmomatic v0.36 (*Bolger et al., 2014*), with a minimum quality score of 10 and minimum read length of 50. Trimmed reads were de novo assembled using SPAdes v3.12.0 (*Bankevich et al., 2012*) with default parameters. Draft assemblies of the 52 NMEC isolates from this study, together with six complete NMEC genomes and eight complete genomes from other characterised *E. coli* representing different phylogroups, were subjected to phylogenetic analysis using parsnp v1.5.3 (*Treangen et al., 2014*). A subset of 18 isolates were additionally sequenced using Oxford Nanopore Technology long-read sequencing (Nanopore). Complete NMEC genomes were achieved using a combination of Illumina short-read and Nanopore long-read data and analysis employing the MicroPIPE tool (*Murigneux et al., 2021*).

### In silico and molecular analyses

Virulence-associated genes, antibiotic resistance genes, plasmids and serotyping were evaluated using ABRicate (RRID:SCR_021093, version 0.8; https://github.com/tseemann/abricate) with built-in databases (*Chen et al., 2016*; *Feldgarden et al., 2019*; *Carattoli et al., 2014*; *Ingle et al., 2016*), with the percentage nucleotide identity and coverage cut-off set at 90% and 80%, respectively. Capsule typing was performed employing Kaptive (*Wyres et al., 2016*) using an in-house *E. coli* capsule database (*Goh et al., 2017*) and manually checked. Chromosomal point mutations associated with antibiotic resistance were detected using PointFinder (*Zankari et al., 2017*). FimH amplicon sequencing was performed as previously described (*Willner et al., 2014*; *Chen et al., 2018*); allelic variants were identified using FimTyper (*Roer et al., 2017*).

### K1 ELISA

K1 capsule expression was detected by ELISA using an anti-polysialic acid antibody single chain Fv fragment (*Nagae et al., 2013*) as the primary antibody, anti-His antibody, and alkaline phosphatase anti-mouse IgG as the secondary and tertiary antibodies, respectively; *p*-nitrophenylphosphate (Sigma) was used as the substrate. Optical density was measured at 420 nm.

### Metagenomic sequencing and analyses

Metagenomic sequencing was performed on DNA extracted from fecal samples using the Illumnina NovaSeq6000 platform. Adapters and low-quality reads were trimmed using Trimmomatic v0.36 (*Bolger et al., 2014*), employing a minimum quality score of 10 and minimum read length of 50. Sequencing reads corresponding to human DNA were discarded by mapping the trimmed reads to the human genome hg38 (accession number GCA_000001405.29) using bowtie2 (*Langmead and Salzberg, 2012*). Taxonomical profiling was performed with Kraken2 (*Wood et al., 2019*) followed by Bracken (*Lu et al., 2017*).

### Long-read metagenomic sequencing

Long-read metagenomic sequencing was performed on DNA extracted from a fecal sample. A HiFi gDNA library was prepared using the SMRTbell Express Template Prep Kit 2.0 (PacBio, 100-938-900) according to the low input protocol (PacBio, PN 101-730-400 Version 06 [June 2020]). As the sample DNA was already fragmented with a tight peak (mode size 9.4 kb), no shearing was performed; the sample was concentrated using Ampure PB beads (PacBio, PCB-100-265-900) and used directly as

input into library preparation. The entire quantity of purified DNA (360 ng) was used to make the library as follows. The DNA was treated to remove single-stranded overhangs, followed by a DNA damage repair reaction and an end-repair/A-tailing reaction. Overhang barcoded adapters were ligated to the A-tailed library fragments, followed by a nuclease treatment to remove damaged library fragments, and then purification with AMPure PB beads. The library was size-selected to remove fragments <3 kb using AMPure PB beads. The final purified, size-selected library was quantified on the Qubit fluorometer using the Qubit dsDNA HS assay kit (Invitrogen, Q32854) to assess concentration, and run on the Agilent Femto Pulse using the 55 kb BAC Analysis Kit (Agilent, FP-1003-0275) to assess fragment size distribution.

Sequencing was performed using the PacBio Sequel II (software/chemistry v10.1). The library pool was prepared for sequencing according to the SMRT Link (v10.1) sample setup calculator, following the standard protocol for Diffusion loading with Ampure PB bead purification, using Sequencing Primer v5, Sequel II Binding Kit v2.2, and the Sequel II DNA Internal Control v1. Adaptive loading was utilised, with nominated on-plate loading concentration of 80 pM. The polymerase-bound library was sequenced on 1 SMRT Cell with a 30-hr movie time plus a 2-hr pre-extension using the Sequel II Sequencing 2.0 Kit (PacBio, 101-820-200) and SMRT Cell 8M (PacBio, 101-389-001).

After sequencing, the data were processed to generate CCS reads and demultiplex samples using the default settings of the CCS with Demultiplexing application in SMRT Link (v10.1). The demultiplexed reads were assembled de novo using Hifiasm (*Cheng et al., 2021*). Assembled contigs were subject to taxonomic profiling using kraken2 (*Wood et al., 2019*) and fastANI (*Jain et al., 2018*). Contigs taxonomically assigned as *E. coli* were subjected to in silico sequence typing using MLST (version 2.11) (https://github.com/tseemann/mlst; *Seemann, 2022*) and mlst profiles from PubMLST (*Jolley and Maiden, 2010*).

## Acknowledgements

The authors would like to thank Michelle Bauer for technical expertise and the laboratories contributing the isolates, Pathology Queensland and Mater Pathology. At the time of the study SS was affiliated with Mater Pathology, South Brisbane, Australia.

## Additional information

### Funding

| Funder | Grant reference number | Author |
| --- | --- | --- |
| National Health and Medical Research Council | APP1181958 | Nguyen Thi Khanh Nhu Minh-Duy Phan Mark A Schembri |
| National Health and Medical Research Council | GNT1197743 | Adam D Irwin |
| Queensland Children's Hospital Foundation | 50270 | Patrick NA Harris Scott A Beatson Sanmarie Schlebusch Adam D Irwin Mark A Schembri |
| Australian Infectious Diseases Research Centre | | Nguyen Thi Khanh Nhu Adam D Irwin Mark A Schembri |
| Wellcome Trust | 10.35802/220211 | Paul Turner |
| National Health and Medical Research Council | APP2001431 | Nguyen Thi Khanh Nhu Minh-Duy Phan Mark A Schembri |

| Funder | Grant reference number | Author |
|---|---|---|

The funders had no role in study design, data collection, and interpretation, or the decision to submit the work for publication. For the purpose of Open Access, the authors have applied a CC BY public copyright license to any Author Accepted Manuscript version arising from this submission.

## Author contributions

Nguyen Thi Khanh Nhu, Formal analysis, Funding acquisition, Investigation, Writing – original draft, Writing – review and editing; Minh-Duy Phan, Formal analysis, Funding acquisition, Investigation, Writing – review and editing; Steven J Hancock, Kate M Peters, Laura Alvarez-Fraga, Brian M Forde, Stacey B Andersen, Thyl Miliya, Investigation, Writing – review and editing; Patrick NA Harris, Scott A Beatson, Sanmarie Schlebusch, Paul Turner, Resources, Funding acquisition, Writing – review and editing; Haakon Bergh, Resources, Writing – review and editing; Annelie Brauner, Benita Westerlund-Wikström, Conceptualization, Resources, Writing – review and editing; Adam D Irwin, Mark A Schembri, Conceptualization, Resources, Formal analysis, Supervision, Funding acquisition, Writing – original draft, Writing – review and editing

## Author ORCIDs

Nguyen Thi Khanh Nhu ⓘ http://orcid.org/0000-0002-0158-850X
Minh-Duy Phan ⓘ http://orcid.org/0000-0002-3426-1044
Patrick NA Harris ⓘ https://orcid.org/0000-0002-2895-0345
Paul Turner ⓘ http://orcid.org/0000-0002-1013-7815
Benita Westerlund-Wikström ⓘ http://orcid.org/0000-0002-5307-3858
Adam D Irwin ⓘ http://orcid.org/0000-0001-8974-6789
Mark A Schembri ⓘ http://orcid.org/0000-0003-4863-9260

## Ethics

The study received ethical approval from the Children's Health Queensland Human Research Ethics Committee (LNR/18/QCHQ/45045), ratified by The University of Queensland (2019000752). The study was deemed as low risk with waiver of informed consent and consent to publish. The Children's Health Queensland Human Research Ethics Committee approved an amendment on 27 July 2021 to report the clinical details included in the case series. Written informed consent was obtained from the carers of each of the infants.

Reviewer #1 (Public Review): https://doi.org/10.7554/eLife.91853.3.sa1
Author response https://doi.org/10.7554/eLife.91853.3.sa2

# Additional files

## Supplementary files

- Supplementary file 1. Isolates used in this study.
- Supplementary file 2. Completely sequenced NMEC isolates.
- Supplementary file 3. Metagenomic sequence analysis.
- Supplementary file 4. Accession numbers of strains sequenced in the study.

## Data availability

Genome sequence data have been deposited in the Sequence Read Archive under the BioProjects PRJNA757133 and PRJNA893826. Sample accession numbers are listed in *Supplementary file 4*.

The following datasets were generated:

| Author(s) | Year | Dataset title | Dataset URL | Database and Identifier |
|---|---|---|---|---|
| Phan M-D, Hancock SJ, Peters KM, Alvarez-Fraga L, Forde BM, Andersen SB, Miliya T, Harris PNA, Beatson SA, Schlebusch S, Bergh H, Turner P, Brauner A, Westerlund-Wikström B, Irwin AD, Schembri MA, Nhu NTK | 2024 | Neonatal meningitis *Escherichia coli* | https://www.ncbi.nlm.nih.gov/bioproject/?term=PRJNA757133 | NCBI BioProject, PRJNA757133 |
| Phan M-D, Hancock SJ, Peters KM, Alvarez-Fraga L, Forde BM, Andersen SB, Miliya T, Harris PNA, Beatson SA, Schlebusch S, Bergh H, Turner P, Brauner A, Westerlund-Wikström B, Irwin AD, Schembri MA, Nhu NTK | 2024 | Culture-independent long read metagenomic diagnostics for genomic surveillance and infection control of pathogenic bacteria in clinical settings | https://www.ncbi.nlm.nih.gov/bioproject/?term=PRJNA893826 | NCBI BioProject, PRJNA893826 |

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
