## [Editor Report · eLife assessment]

This **valuable** study presents findings characterising the genomic features of *E. coli* isolated from neonatal meningitis from seven countries, and documents bacterial persistence and reinfection in two case studies. The genomic analyses are **solid**, although the inclusion of a larger number of isolates from more diverse geographies would have strengthened the generalisability of findings. The work will be of interest to people involved in the management of neonatal meningitis patients, and those studying *E. coli* epidemiology, diversity, and pathogenesis.

---

## [Referee Report · Reviewer #1 (Public Review)]

Summary:

This study uses whole genome sequencing to characterise the population structure and genetic diversity of a collection of 58 isolates of *E. coli* associated with neonatal meningitis (NMEC) from seven countries, including 52 isolates that the authors sequenced themselves and a further 6 publicly available genome sequences. Additionally, the study used sequencing to investigate three case studies of apparent relapse. The data show that in all three cases, the relapse was caused by the same NMEC strain as the initial infection. In two cases they also found evidence for gut persistence of the NMEC strain, which may act as a reservoir for persistence and reinfection in neonates. This finding is of clinical importance as it suggests that decolonisation of the gut could be helpful in preventing relapse of meningitis in NMEC patients.

Strengths:

The study presents complete genome sequences for n=18 diverse isolates, which will serve as useful references for future studies of NMEC. The genomic analyses are high quality, the population genomic analyses are comprehensive and the case study investigations are convincing. The full data set (including phylogenetic tree, annotated with source, lineage and virulence factor information) are publicly available in interactive form via the MicroReact platform.

Weaknesses:

The NMEC collection described in the study includes isolates from just seven countries. The majority (n=51/58, 88%) are from high-income countries in Europe, Australia or North America; the rest are from Cambodia (n=7, 12%). Therefore it is not clear how well the results reflect the global diversity of NMEC, nor the populations of NMEC affecting the most populous regions.

The virulence factors section highlights several potentially interesting genes that are present at apparently high frequency in the NMEC genomes; however without knowing their frequency in the broader *E. coli* population it is hard to know the significance of this.

---

## [Author Response]

The following is the authors’ response to the original reviews.

**eLife assessment**
This study presents valuable findings characterising the genomic features of *E. coli* isolated from neonatal meningitis from seven countries, and documents bacterial persistence and reinfection in two case studies. The genomic analyses are solid, although the inclusion of a larger number of isolates from more diverse geographies would have strengthened the generalisability of findings. The work will be of interest to people involved in the management of neonatal meningitis patients, and those studying *E. coli* epidemiology, diversity, and pathogenesis.
**Public Reviews:**

**Reviewer #1 (Public Review):**
Summary:This study uses whole genome sequencing to characterise the population structure and genetic diversity of a collection of 58 isolates of *E. coli* associated with neonatal meningitis (NMEC) from seven countries, including 52 isolates that the authors sequenced themselves and a further 6 publicly available genome sequences. Additionally, the study used sequencing to investigate three case studies of apparent relapse. The data show that in all three cases, the relapse was caused by the same NMEC strain as the initial infection. In two cases they also found evidence for gut persistence of the NMEC strain, which may act as a reservoir for persistence and reinfection in neonates. This finding is of clinical importance as it suggests that decolonisation of the gut could be helpful in preventing relapse of meningitis in NMEC patients.Strengths:The study presents complete genome sequences for n=18 diverse isolates, which will serve as useful references for future studies of NMEC. The genomic analyses are high quality, the population genomic analyses are comprehensive and the case study investigations are convincing.

We agree

Weaknesses:The NMEC collection described in the study includes isolates from just seven countries. The majority (n=51/58, 88%) are from high-income countries in Europe, Australia, or North America; the rest are from Cambodia (n=7, 12%). Therefore it is not clear how well the results reflect the global diversity of NMEC, nor the populations of NMEC affecting the most populous regions.The virulence factors section highlights several potentially interesting genes that are present at apparently high frequency in the NMEC genomes; however, without knowing their frequency in the broader *E. coli* population it is hard to know the significance of this.

We acknowledged the limitations of our NMEC collection in the Discussion. We agree the prevalence of virulence factors in our collection is interesting. The limited size of our collection prevented further evaluation of the prevalence of these virulence factors in a broader *E. coli* population.

**Reviewer #2 (Public Review):**
Summary:In this work, the authors present a robust genomic dataset profiling 58 isolates of neonatal meningitis-causing *E. coli* (NMEC), the largest such cohort to be profiled to date. The authors provide genomic information on virulence and antibiotic resistance genomic markers, as well as serotype and capsule information. They go on to probe three cases in which infants presented with recurrent febrile infection and meningitis and provide evidence indicating that the original isolate is likely causing the second infection and that an asymptomatic reservoir exists in the gut. Accompanying these results, the authors demonstrate that gut dysbiosis coincides with the meningitis.Strengths:The genomics work is meticulously done, utilizing long-read sequencing.The cohort of isolates is the largest to be sampled to date.The findings are significant, illuminating the presence of a gut reservoir in infants with repeating infection.

We agree

Weaknesses:Although the cohort of isolates is large, there is no global representation, entirely omitting Africa and the Americas. This is acknowledged by the group in the discussion, however, it would make the study much more compelling if there was global representation.

We agree. In the Discussion we state this is likely a reflection of the difficulty in acquiring isolates causing neonatal meningitis, in particular from countries with limited microbiology and pathology resources.

**Reviewer #3 (Public Review):**
Summary:In this manuscript, Schembri et al performed a molecular analysis by WGS of 52 *E. coli* strains identified as "causing neonatal meningitis" from several countries and isolated from 1974 to 2020. Sequence types, virulence genes content as well as antibiotic-resistant genes are depicted. In the second part, they also described three cases of relapse and analysed their respective strains as well as the microbiome of three neonates during their relapse. For one patient the same *E. coli* strain was found in blood and stool (this patient had no meningitis). For two patients microbiome analysis revealed a severe dysbiosis.Major comments:Although the authors announce in their title that they study *E. coli* that cause neonatal meningitis and in methods stipulate that they had a collection of 52 NMEC, we found in Supplementary Table 1, 29 strains (therefore most of the strains) isolated from blood and not CSF. This is a major limitation since only strains isolated from CSF can be designated with certainty as NMEC even if a pleiocytose is observed in the CSF. A very troubling data is the description of patient two with a relapse infection. As stated in the text line 225, CSF microscopy was normal and culture was negative for this patient! Therefore it is clear that patient without meningitis has been included in this study.

We have reviewed the clinical data for our 52 NMEC isolates, noting that for some of the older Finish isolates we relied on previous publications. This data is shown in Table S1. To address the Reviewer’s comment, we have added the following text to the methods section (new text underlined).

‘The collection comprised 42 isolates from confirmed meningitis cases (29 cultured from CSF and 13 cultured from blood) and 10 isolates from clinically diagnosed meningitis cases (all cultured from blood).’

Patient 2 was initially diagnosed with meningitis based on a positive blood culture in the presence of CSF pleocytosis (>300 WBCs, >95% polymorphs). We understand there may be some confusion with reference to a relapsed infection, which we now more accurately describe as recrudescent invasive infection in the revised manuscript.

Another major limitation (not stated in the discussion) is the absence of clinical information on neonates especially the weeks of gestation. It is well known that the risk of infection is dramatically increased in preterm neonates due to their immature immunity. Therefore *E. coli* causing infection in preterm neonates are not comparable to those causing infection in term neonates notably in their virulence gene content. Indeed, it is mentioned that at least eight strains did not possess a capsule, we can speculate that neonates were preterm, but this information is lacking. The ages of neonates are also lacking. The possible source of infection is not mentioned, notably urinary tract infection. This may have also an impact on the content of VF.

We agree. In the Discussion we now note the following (new text underlined):

‘… we did not have clinical data on the weeks of gestation for all patients, and thus could not compare virulence factors from NMEC isolated from preterm versus term infants.’

Submission to Medrxiv, a requirement for review of our manuscript at eLife, necessitated the removal of some patient identifying information, including precise age and detailed medical history.

Sequence analysis reveals the predominance of ST95 and ST1193 in this collection. The high incidence of ST95 is not surprising and well previously described, therefore, the concluding sentence line 132 indicating that ST95 *E. coli* should exhibit specific virulence features associated with their capacity to cause NM does not add anything. On the contrary, the high incidence of ST1193 is of interest and should have been discussed more in detail. Which specific virulence factors do they harbor? Any hypothesis explaining their emergence in neonates?

We compared the virulence factors of ST95 and ST1193 and summarized this information in Figure 4. We also discussed how the K1 polysialic acid capsule in ST95 and ST1193 could contribute to the emergence of these STs in NM. Specifically, we stated the following: ‘We speculate this is due to the prevailing K1 polysialic acid capsule serotype found in ST95 and the newly emerged ST1193 clone [22, 37] in combination with other virulence factors [15, 28, 29] (Figure 4) and the immature immune system of preterm infants.’

In the paragraph depicted the VF it is only stated that ST95 contained significantly more VF than the ST1193 strains. And so what? By the way "significantly" is not documented: n=?, p=?

We compared the prevalence of known virulence factors between ST95 and ST1193, and showed that ST95 strains in our collection contained significantly more virulence factors than the ST1193 strains. The P-value and the statistical test used were included in Supplementary Figure 3. To address the reviewers concern, we have now also added this to the main manuscript text as follows (new text underlined):

‘Direct comparison of virulence factors between ST95 and ST1193, the two most dominant NMEC STs, revealed that the ST95 isolates (n = 20) contained significantly more virulence factors than the ST1193 isolates (n=9), p-value < 0.001, Mann-Whitney two-tailed unpaired test (Supplementary Table 1, Supplementary Figure 3).’

The complete sequence of 18 strains is not clear. Results of Supplementary Table 2 are presented in the text and are not discussed.

NMEC isolates that were completely sequenced in this study are indicated in bold and marked with an asterisk in Figure 1. This information is indicated in the figure legend and was provided in the original submission. All information regarding genomic island composition and location, virulence genes and plasmid and prophage diversity is included in Supplementary Table 2. This information is highly descriptive and thus we elected not to include it as text in the main manuscript.

46 years is a very long time for such a small number of strains, making it difficult to put forward epidemiological or evolutionary theories. In the analysis of antibiotic resistance, there are no ESBLs. However, Ding's article (reference 34) and other authors showed that ESBLs are emerging in *E. coli* neonatal infection. These strains are a major threat that should be studied, unfortunately, the authors haven't had the opportunity to characterize such strains in their manuscript.

We agree 46 years is a long time-span. The study by Ding et al examined 56 isolates comprised of 25 different STs isolated in China from 2009-2015, with ST1193 (n=12) and ST95 (n=10) the most common. Our study examined 58 isolates comprised of 22 different STs isolated in seven different geographic regions from 1974-2020, with ST1193 (n=9) and ST95 (n=20) the most common. Thus, despite differences in the geographic regions from which isolates in the two studies were sourced, there are similarities in the most common STs identified. The fact that we observed less antibiotic resistance, including a lack of ESBL genes, in ST1193 is likely due to the different regions from which the isolates were sourced. We acknowledged and discussed the potential of ST1193 harbouring multidrug resistance including ESBLs in our manuscript as follows:

‘Concerningly, the ST1193 strains examined here carry genes encoding several aminoglycoside-modifying enzymes, generating a resistance profile that may lead to the clinical failure of empiric regimens such as ampicillin and gentamicin, a therapeutic combination used in many settings to treat NM and early-onset sepsis [35, 36]. This, in combination with reports of co-resistance to third-generation cephalosporins for some ST1193 strains [22, 34], would limit the choice of antibiotic treatment.’

Second part of the manuscript:The three patients who relapsed had a late neonatal infection (> 3 days) with respective ages of 6 days, 7 weeks, and 3 weeks. We do not know whether they are former preterm newborns (no term specified) or whether they have received antibiotics in the meantime.

As noted above, patient ages were not disclosed to comply with submission to Medrxiv, a requirement for review of our manuscript at eLife.

Patient 1: Although this patient had a pleiocytose in CSF, the culture was negative which is surprising and no explanation is provided. Therefore, the diagnosis of meningitis is not certain. Pleiocytose without meningitis has been previously described in neonates with severe sepsis. Line 215: no immunological abnormalities were identified (no details are given).

We respectfully disagree with the reviewer. The diagnosis of meningitis is made unequivocally by the presence of a clearly abnormal CSF microscopy (2430 WBCs) and an invasive *E. coli* from blood culture. This does not seem controversial to the authors. We had believed it unnecessary to include this corroborative evidence, but have added the following to support our assertion:

‘The child was diagnosed with meningitis based on a cerebrospinal fluid (CSF) pleocytosis (>2000 white blood cells; WBCs, low glucose, elevated protein), positive CSF *E. coli* PCR and a positive blood culture for *E. coli* (MS21522).’

On the contrary, the authors are surprised by the statement that CSF pleocytosis occurs in neonatal sepsis ‘without meningitis’ and do not know of any definitions of neonatal meningitis that are not tied to the presence of a CSF pleocytosis. Furthermore, the later isolation of *E. coli* from the CSF during the relapsed infection re-enforces the initial diagnosis.

Patient 2: This patient had a recurrence of bacteremia without meningitis (line 225: CSF microscopy was normal and culture negative!). This case should be deleted.

In a similar vein to the previous comment, we respectfully assert that this patient has clear evidence of meningitis (330 WBCs in the CSF, taken 24h after initiation of antibiotic treatment). In this case, molecular testing was not performed as, under the principle of diagnostic stewardship, it was not considered necessary by the clinical microbiologists and treating clinicians following the culture of *E. coli* in the bloodstream. We agree that this is not a case of recurrent meningitis, but our intention was to highlight the recrudescence of an invasive infection (urinary sepsis requiring admission to hospital and intravenous antibiotics) which we hypothesise has arisen from the intestinal reservoir. We did not state that all patients suffered from relapsed meningitis.

Despite this, to address this reviewers concern, we have changed all reference to ‘relapsed infection’ to now read ‘recrudescent invasive infection’ in the revised manuscript.

Patient 3: This patient had two relapses which is exceptional and may suggest the existence of a congenital malformation or a neurological complication such as abscess or empyema therefore, "imaging studies" should be detailed.

This patient underwent extensive imaging investigation to rule out a hidden source. This included repeated MRI imaging of head and spine, CT imaging of head and chest, USS imaging of abdomen and pelvis and nuclear medicine imaging to detect a subtle meningeal defect and CSF leak. All tests were normal, and no abscess or empyema found.

We have modified the text to include this information:

Text in original submission: ‘Imaging studies and immunological work-up were normal.’

New text in revised manuscript (underlined): ‘Extensive imaging studies including repeated MRI imaging of the head and spine, CT imaging of the head and chest, ultrasound imaging of abdomen and pelvis, and nuclear medicine imaging did not show a congenital malformation or abscess. Immunological work-up did not show a known primary immunodeficiency. At two years of age, speech delay is reported but no other developmental abnormality.’

The authors suggest a link between intestinal dysbiosis and relapse in three patients. However, the fecal microbiomes of patients without relapse were not analysed, so no comparison is possible.Moreover, dysbiosis after several weeks of antibiotic treatment in a patient hospitalized for a long time is not unexpected. Therefore, it's impossible to make any assumption or draw any conclusion. This part of the manuscript is purely descriptive. Finally, the authors should be more prudent when they state in line 289 "we also provide direct evidence to implicate the gut as a reservoir [...] antibiotic treatment". Indeed the gut colonization of the mothers with the same strain may be also a reservoir (as stated in the discussion line 336). Finally, the authors do not discuss the potential role of ceftriaxone vs cefotaxime in the dysbiosis observed. Ceftriaxone may have a major impact on the microbiota due to its digestive elimination.

We addressed the limitations of our study in the Discussion, including that we did not have access to urine or stool samples from the mother of the infants that suffered recrudescence, and thus cannot rule out mother-to-child transmission as a mechanism of reinfection. We have now added that we did not have clinical data on the weeks of gestation for all patients, and thus could not compare virulence factors from NMEC isolated from preterm versus term infants. The limitations of our study are summarised as follows in the Discussion (new text underlined):

‘This study had several limitations. First, our NMEC strain collection was restricted to seven geographic regions, a reflection of the difficulty in acquiring strains causing this disease. Second, we did not have access to a complete set of stool samples spanning pre- and post-treatment in the patients that suffered NM and recrudescent invasive infection. This impacted our capacity to monitor *E. coli* persistence and evaluate the effect of antibiotic treatment on changes in the microbiome over time. Third, we did not have access to urine or stool samples from the mother of the infants that suffered recrudescence, and thus cannot rule out mother-to-child transmission as a mechanism of reinfection. Finally, we did not have clinical data on the weeks of gestation for all patients, and thus could not compare virulence factors from NMEC isolated from preterm versus term infants.’

**Recommendations for the authors:**

**Reviewer #1 (Recommendations For The Authors):**
It would be useful to mention the sample size (number of genomes analysed, n=58) in the abstract to give readers a sense of the scale of the analysis.

We have added the number of genomes in the abstract as suggested (new text underlined).

‘Here we investigated the genomic relatedness of a collection of 58 NMEC strains spanning 1974-2020 and isolated from seven different geographic regions.’

The term 'strain' is used throughout, it would be clearer to use 'isolates' to describe the biological material and 'genomes' when the unit being referred to is genome sequences. For example, lines 108-111 use 'strain' to mean the collection of 52 isolates but also uses 'strain' to mean the collection of 58 genomes including those of the 52 isolates that the authors sequenced plus a further 6 genomes of isolates that they do not have in their isolate collection.

We have changed the term ‘strain’ to ‘isolate’ or ‘genome’ as suggested.

Figure 1 (annotated phylogeny) is hard to read and interpret, as so much data is presented. It would assist readers if the authors could provide an interactive form of the phylogeny and metadata/genomic feature data discussed in the text, e.g. using microreact.org, so that details can be explored more easily.

This is an excellent suggestion, and we created a project on microreact.org. This information has been added to the Figure 1 legend.

https://microreact.org/project/oNfA4v16h3tQbqREoYtCXj-high-risk-escherichia-coli-clones-that-cause-neonatal-meningitis-and-association-with-recrudescent-infection.

It would be useful to provide information on the frequency and/or distribution of the virulence factors in the broader *E. coli* population, to provide context for readers and to better understand the importance/significance of the high frequency of the reported virulence factors within NMEC.

As noted above, we agree the prevalence of virulence factors in our collection is interesting. We discussed the prevalence of these virulence factors in our collection, and the detailed data is presented in Table S1. However, we also note a limitation in our study is the number of isolates, and thus we would prefer to avoid evaluation of the prevalence of these virulence factors in the context of a broader *E. coli* population. There are other studies that have examined NMEC virulence factors in the past; some examples are noted below, and we have now referenced these in our manuscript (note Ref 15 was suggested by Reviewer 3 in a comment below; PMID: 11920295).

Ref 15: Johnson JR, Oswald E, O'Bryan TT, Kuskowski MA, Spanjaard L. Phylogenetic distribution of virulence-associated genes among *Escherichia coli* isolates associated with neonatal bacterial meningitis in the Netherlands. J Infect Dis 2002; 185(6): 774-84.

Ref 28: Wijetunge DS, Gongati S, DebRoy C, et al. Characterizing the pathotype of neonatal meningitis causing *Escherichia coli* (NMEC). BMC Microbiol 2015; 15: 211.

Ref 29: Bidet P, Mahjoub-Messai F, Blanco J, et al. Combined Multilocus Sequence Typing and O Serogrouping Distinguishes *Escherichia coli* Subtypes Associated with Infant Urosepsis and/or Meningitis. J Infect Dis. 2007; 196(2):297-303.

I suggest avoiding the term 'global' to describe the collection, given that there are only seven countries included in the collection and two of the most populous continents (Africa and South America) are not represented at all.

We agree, and now refer to our collection as ‘an NMEC strain collection from geographically diverse locations.’

**Reviewer #2 (Recommendations For The Authors):**
This is a suggestion regarding discussion/food for thought: This study sheds information on genomic features and indicates the presence of a reservoir in the infected infant. Previous studies have demonstrated the presence of a reservoir in the vaginas of women with recurrent UTIs. Is there any information as to whether the mothers of these infants, especially the three with recrudescent infection, had a UTI or recurrent UTI in their life? It may be worthwhile discussing the potential of testing for *E. coli* in expecting mothers, if they have a history of UTI.

We do not have such data, and as indicated above we note this as a limitation of our study.

It is unclear as written in the main text, as to whether all three cases of recrudescent infection come from the same geographical location. It would be easier to have this information in the corresponding main text, in addition to the supplement.

The three cases of recrudescent invasive infection were from 3 different locations. We have added the information as following (new text underlined):

‘These patients were from different regions in Australia.’

**Reviewer #3 (Recommendations For The Authors):**
Line 48 and 67 change the word "devasting".

Changed as suggested.

Line 49 second most in full-term infants.

Changed as suggested.

Line 56 delete the sentence "antibiotic resistance genes occurred infrequently".

We changed the sentence, which now reads (new text underlined):

‘Antibiotic resistance genes occurred infrequently in our collection’.

Line 76 reference 10 is inappropriate.

Reference 10 reported that 5/24 infants treated for neonatal Gram-negative bacillary meningitis over a 10-year period had a relapse of meningitis after the initial course of treatment. Four of the isolates that caused these relapsed infections were *E. coli.*

To address the reviewers concern, we have altered the text as follows (new text underlined):

‘Moreover, NMEC is an important cause of relapsed infections in neonates [10]’.

Line 83 several references related to serotypes are missing, notably doi.org/10.1086/339343.

We have added this reference.

Line 171 significantly? n=?, p=?

The numbers and P-value were provided in the Supplementary Figure 3 legend. We have now added these to the text as follows:

‘Direct comparison of virulence factors between ST95 and ST1193, the two most dominant NMEC STs, revealed that the ST95 isolates (n = 20) contained significantly more virulence factors than the ST1193 isolates (n = 9); P-value < 0.001, Mann-Whitney two-tailed unpaired test (Supplementary Table 1, Supplementary Figure 3).”

Figure 4 is not necessary.

We respectfully disagree. Figure 4 provides an illustrative comparison of virulence factors between the two most dominant NMEC sequence types, ST95 and ST1193. We believe this will be informative for many readers.

Line 311 "We speculate....of preterm infants" This sentence does not add anything to the discussion.

We respectfully disagree and have kept the sentence. This reflects our opinion.

Line 320 "clear clinical risk factors to explain... ». Term of neonates is missing.

Updated as follows (new text underlined):

‘Although reported rarely, recrudescent invasive *E. coli* infection in NM patients, including several infants born pre-term, has been documented in single study reports [39, 40]. In these reports, infants received appropriate antibiotic treatment based on antibiogram profiling and no clear clinical risk factors to explain recrudescence were identified, highlighting our limited understanding of NM aetiology.’